# Matrigel 3D bioprinting of contractile human skeletal muscle models recapitulating exercise and pharmacological responses

Angela Alave Reyes-Furrer[1], Sonia De Andrade [1,2], Dominic Bachmann[1], Heidi Jeker[1], Martin Steinmann[3], Nathalie Accart[1], Andrew Dunbar[3], Martin Rausch[3], Epifania Bono[2,4], Markus Rimann [2,4✉] & Hansjoerg Keller [1✉]

A key to enhance the low translatability of preclinical drug discovery are in vitro human three-dimensional (3D) microphysiological systems (MPS). Here, we show a new method for automated engineering of 3D human skeletal muscle models in microplates and functional compound screening to address the lack of muscle wasting disease medication. To this end, we adapted our recently described 24-well plate 3D bioprinting platform with a printhead cooling system to allow microvalve-based drop-on-demand printing of cell-laden Matrigel containing primary human muscle precursor cells. Mini skeletal muscle models develop within a week exhibiting contractile, striated myofibers aligned between two attachment posts. As an in vitro exercise model, repeated high impact stimulation of contractions for 3 h by a custom-made electrical pulse stimulation (EPS) system for 24-well plates induced *interleukin-6* myokine expression and Akt hypertrophy pathway activation. Furthermore, the known muscle stimulators caffeine and Tirasemtiv acutely increase EPS-induced contractile force of the models. This validated new human muscle MPS will benefit development of drugs against muscle wasting diseases. Moreover, our Matrigel 3D bioprinting platform will allow engineering of non-self-organizing complex human 3D MPS.

[1] Musculoskeletal Disease Area, Novartis Institutes for BioMedical Research, Basel, Switzerland. [2] 3D Tissues and Biofabrication, Institute of Chemistry and Biotechnology (ICBT), Zurich University of Applied Sciences, Waedenswil, Switzerland. [3] Analytical Sciences & Imaging, Novartis Institutes for BioMedical Research, Basel, Switzerland. [4] Competence Center TEDD, Institute of Chemistry and Biotechnology (ICBT), Zurich University of Applied Sciences, Waedenswil, Switzerland. ✉email: markus.rimann@zhaw.ch; hansjoerg.keller@novartis.com

Skeletal muscle is the most abundant tissue and organ of the body by mass covering about 20–50% of the human body weight[1]. It provides independent, free movement of the body and quick reaction to external stimuli. In our aging societies, musculoskeletal conditions are on a steep rise and are projected to become the most abundant diseases, because prevalent maladies such as cancer and cardiovascular disorders will regress due to healthy life-style, disease prevention efforts and continuously emerging new improved medication[2]. Loss of mobility by muscle wasting conditions leads to huge health care costs including wheel chair assistance and accessible home, as well as assistance personnel at home, in hospitals or nursing homes[3]. Strikingly, despite this great medical need, there is a complete lack of disease-modifying medication[4]. First emerging drug therapies in clinical trials target, for example, the androgen or myostatin pathways, which control skeletal muscle mass[5–7]. However, clinical results were mixed so far showing muscle mass increases but no conclusive muscle function improvements.

Drug development is a lengthy and costly process taking on average about 15 years and requiring about 2.5 billion dollars. One of the main reasons for this is the very low success rate of new drug candidates entering clinical testing. Despite great advances on functional genomics, disease understanding, virtual modelings and computational simulations, in vitro high throughput screenings and animal experimentations, 9 out of 10 drug candidates fail in the clinic due to still not-foreseen liabilities such as insufficient activity and intolerable safety profile[8]. A key to enhance the translatability of preclinical drug discovery and development into humans are in vitro human 3D microphysiological systems (MPS), which mimic aspects of human physiology and disease in a dish. Thus, MPS promise more translatable compound screens that result in drug candidates with higher success rates in the clinic[9]. MPS not only allow drug candidate testing on human tissue models, but also on personalized patient-derived disease tissue models using modern stem cell re-programming and tissue engineering tools such as induced pluripotent stem cell and CRISPR technologies[9].

Skeletal muscle is one of the human tissues with life-long growth and regeneration capability due to the presence of dormant stem cells, which can be activated to build or re-build skeletal muscle. Since many years, muscle fiber development can be reproduced in vitro using the precursor cells isolated from skeletal muscle biopsies. Such 2D tissue culture assays have been used to identify drug candidates regulating growth and maintenance of skeletal muscle such as myostatin pathway blockers[10]. However, these assays do not allow screening for drug candidates regulating core muscle function such as contractile force and fatigue. Thus, 3D in vitro tissue-like models of contracting skeletal muscle fibers were developed for the screening of compounds regulating contractility[11,12]. These human tissue models mimicked pharmacological responses of drugs used in the clinic[13]. However, so far none of these models was reliable and robust enough for routine use in drug screening. Recently, we developed a 3D bioprinting platform for the automated engineering of human skeletal muscle tissue models in multiwell plates[14]. Dumbbell-shaped 3D micro tissue models were reliably printed in 24-well plates in a combination of microvalve-based drop-on-demand (DOD) muscle cells printing[14,15] with alternating layers of a PEG-based photo-polymerizable bioink in extrusion mode. However, cultivated models showed poor myofiber differentiation with minimal contractility, because the applied bioink was apparently metabolically inert and did not support tissue development and functional maturation.

Matrigel is a commonly used extract of extracellular basement membrane proteins derived from the Engelbreth-Holm-Swarm mouse sarcoma to support growth and differentiation of cells in culture[16–18]. Despite its tumor origin, which prevents its clinical application, it is still used for many in vitro tissue engineered models because of missing alternatives providing similar biological cues. Recently, it also became a frequently used key component of the emerging organoid technology, which generates complex human tissues in a dish from stem cells[19]. Likewise, it has been widely used together with fibrin as a composite hydrogel for the generation of 3D skeletal muscle tissue models[12]. Despite these successes in 3D tissue culture, Matrigel was rarely used as a bioink and mostly combined with other materials to improve its printability with extrusion bioprinters[20–23].

In our previous work, we printed co-cultures of tenocytes and myoblasts to generate complex muscle-tendon tissue models jetting cells on the top of polymerized bioinks[14]. The microvalve-based DOD 3D bioprinting technique allows a controlled spatial deposition of cells, which is not possible with other standard techniques such as micro-molding[24]. To remain flexible in terms of tissue complexity, we continued using the same bioprinting technique. Here, we describe the establishment of microvalve-based DOD 3D bioprinting of pure Matrigel suspensions containing human skeletal muscle precursor cells using a cooled printhead in an automated multiwell plate 3D bioprinting platform. Dumbbell-shaped printed cell/Matrigel suspensions solidified in the microplate wells at room temperature followed by 37 °C. Skeletal muscle myofiber tissues developed between two inserted posts after culturing for several days. EPS-induced contractions that were blocked by the myosin inhibitor blebbistatin and by the voltage-gated sodium channel blocker tetrodotoxin (TTX) indicating physiological excitation-contraction coupling. In addition, exercise-mimicking repeated EPS induced Akt phosphorylation and interleukin-6 (IL-6) myokine expression. Finally, contractile force measurements showed significant acute stimulatory effects of the known muscle force enhancers caffeine and Tirasemtiv.

## Results

**3D bioprinting of cell-laden Matrigel models**. To establish microvalve-based DOD 3D bioprinting of cells suspended in liquid pure Matrigel, a suitable cooling system was developed for the printheads and cartridges of a 3D bioprinter (Fig. 1a), since Matrigel solutions are liquid at low temperature (<10 °C) and solidify at 24–37 °C within 30 min[16,17]. The circulating water-based cooling system consisted of insulated tubing connected to a water bath that is temperature controlled by an integrated temperature sensor placed in the cooling jacket of the print cartridge. We used a set temperature of 7 °C for the cooling jacket. The temperature was verified with an external contact thermometer after an equilibration time of >40 min measuring 7.3 ± 0.14 °C at the printhead. Higher temperatures of 13.4 ± 0.22 °C were detected at the microvalve orifice below the cooling jacket (Fig. 1a). These temperatures remained constant over >2 h-printing time.

In order to assess the printability of cooled cell-laden Matrigel bioink by microvalve-based mode and subsequent gelling in 24-well plates at RT followed by 37 °C, we first analyzed the rheological properties of different Matrigel protein concentrations and batches on ice and their gelling at 37 °C. Two different batches of Matrigel with similar total protein concentrations of 8.1 and 9.5 mg/mL, respectively, were tested undiluted or diluted as specified in Fig. 1b. Droplets (50 μL) of chilled Matrigel solutions were manually pipetted onto glass slides followed by incubation at 37 °C for 30 min. As expected, we observed a positive correlation between protein concentration and solution viscosity. After solidification at 37 °C, droplets with a concentration of >3 mg/mL were solid, but a concentration of >4 mg/mL was required to allow model lifting-up with a spatula and keeping

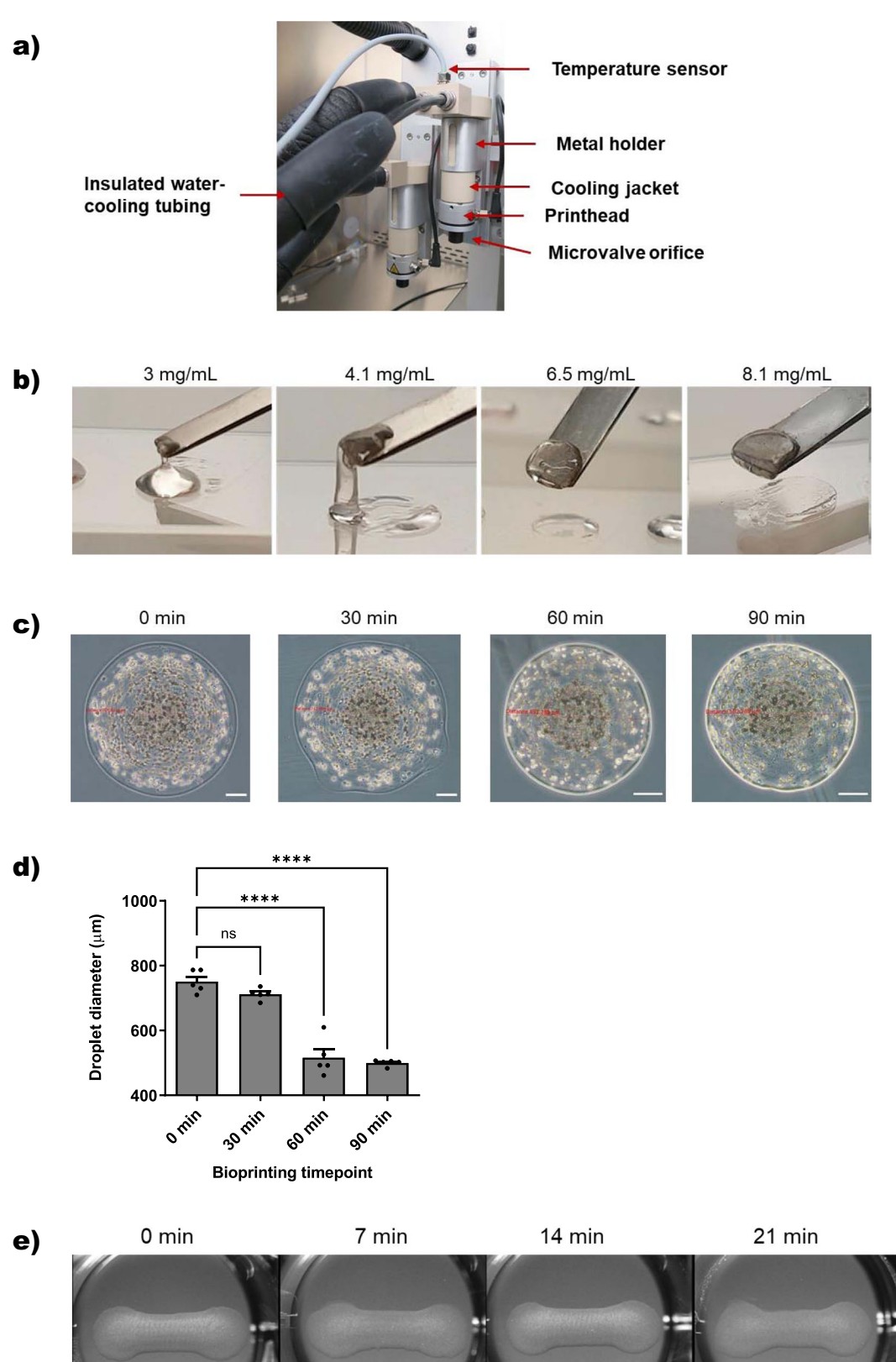

the shape (Fig. 1b). This concentration correlation was independent of the used batch. Undiluted Matrigel solutions of 8.1 and 9.5 mg/mL total protein concentration showed best shape retention and good rheological properties for pipetting at 0–7 °C. Therefore, we choose pure Matrigel solution as cell-laden bioink for microvalve DOD 3D bioprinting.

Primary proliferating human muscle precursor cells were suspended at a density of $2 \times 10^7$ cells/mL in pure Matrigel solutions on ice immediately before 3D bioprinting. Four-layer dumbbell-shaped 3D models were printed on agarose substrates in 24-well plates similarly as recently reported[14]. Previously, we developed a cell-stirring system to avoid cell sedimentation in the

**Fig. 1 Microvalve-based DOD 3D bioprinting of pure Matrigel/cell suspensions. a** Images of the printhead cooling system. Two printheads were equipped with cooling jackets for the printing cartridges. Insulated tubing connected to a temperature-controlled water bath delivered circulating water-based cooling. An integrated temperature sensor in the cooling jacket provided temperature control. **b** Qualitative spatula lift-up gelling tests of Matrigel drops with increasing total protein contents as indicated in the figure. **c** Microscopic images of printed Matrigel/cell suspension droplets over time. Scale bar 100 μm. **d** Quantification of printed droplet diameter ($n = 5$, means ± sem). Statistics: Ordinary one-way ANOVA, Bonferroni's multiple comparison test, ****$p < 0.0001$. **e** Macroscopic images of dumbbell-shaped Matrigel/cell models directly after printing at different time points of the printing process in 24-well plates on agarose substrates. Scale bar 2 mm.

print cartridges during printing in order to achieve long-term printing of constant cell concentrations[14]. Since Matrigel is more viscous than aqueous cell culture media, we estimated that cells do not sediment considerably in Matrigel suspensions. Thus, we did not use the cell stirring system to maintain a homogenous cell suspension in this current project. Indeed, as shown in Fig. 1c, single droplets of Matrigel/cell suspensions of comparable size and apparent similar cell content were printed for up to 90 min showing only a 30% decrease in diameter size starting with droplets printed 60 min after cartridge loading (Fig. 1d). Accordingly, 3D models were printed with high reproducibility as illustrated with four representative models in Fig. 1e. Microvalves could be re-used for printing after thorough cleaning. Impure microvalves, as observed by macroscopically inspection of the transparent orifice showing internal debris, resulted in frequent formation of a large drop at the orifice during the print process rendering further printing impossible (Supplementary Fig. 1). In summary, we have developed reliable 3D bioprinting of cooled Matrigel/cell suspensions on agarose substrate in 24-well plates at RT.

**Skeletal muscle tissue model differentiation.** Skeletal muscle tissue is composed of bundles of multinucleated myofibers. They form by fusion of precursor cells during muscle organogenesis in the embryo as well as during muscle growth and regeneration in the adult organism. Subsequently, maturation of the myofiber contractile apparatus and development of longitudinal tensile force require co-development of tendon/bone attachment points of matched strength for proper muscle function. Recently, we have described novel cell culture inserts with two posts for the attachment and maturation of differentiating human myofibers in 24-well plates[14]. However, the posts were too stiff, not allowing dynamic contractions of the 3D bioprinted myofiber/bioink (gelatine methacryloyl-polyethylene glycol dimethacrylate [GelMA-PEGDA]) sandwich models by post bending, which resulted in poor tissue differentiation and frequent tearing of the muscle models. Similarly, Matrigel/cell models printed on these postholder inserts mostly ruptured around differentiation day 4 without any observations of post bending. Therefore, as a technically feasible and more flexible post alternative, we printed the models into wells without postholder inserts and simply manually inserted thin portions of pipette tips as posts into the agarose substrate at the designated post locations. We hypothesized that the more flexible posts would better match tensile force development of maturing myofibers. Indeed, most of the models developed well, remained attached between the two inserted posts and did not rupture until day 8 (Fig. 2a). During this time, models condensed losing about 50% of width in the middle between the posts. In contrast, printed models without inserted posts, but immobilized with a Matrigel overlay, condensed less in width, but clearly shrunk in length (Fig. 2b). These results illustrate longitudinal tensile force production by differentiating myofibers in the models. Finally, we analyzed the development of models from different donors with posts and over longer time periods (Fig. 2c). Models from 19- and 40-year-old donors mostly

remained attached to the posts for more than 3 weeks, whereas models from a 17-year-old donor all ruptured after 10 days.

Next, we analyzed tissue differentiation of the models by marker gene analysis (Fig. 3a). We compared differentiation of muscle precursor cells from a 17-year-old donor in 2D culture to 3D bioprinted models and to 3D models from a 19-year-old donor. As expected, the myoblast proliferation marker gene *Myf5* was strongly downregulated both in 2D and 3D cultures within the first week when myoblasts exit cell cycle, fuse and form visible myotubes. Conversely, myogenic differentiation marker genes such as *MyoD* and *Myog* steeply rose during differentiation and much more in 3D models compared to 2D cultures. Furthermore, 3D models expressed structural genes of the myofiber contractile apparatus (*Actn2* and *Myhs*) in general much more strongly and over longer time than 2D cultures. Between the two donors, 3D models from the 19-year-old donor showed a particularly high and sustained expression of these markers indicating stronger tissue differentiation and maturation. This fits with the markedly longer lifetime of 3D models from this donor compared to the 17-year-old donor (Fig. 2c). Myosin heavy chain (*Myh*) subtype analyses showed highest expression of slow fiber type *Myh7* followed by embryonal *Myh3* and postnatal *Myh8*. In contrast, the fast fiber-types *Myh1* & *2* were only marginally expressed. This *Myh* expression pattern indicated that the in vitro engineered 3D models from 19-year-old donor mainly consisted of slow and embryonal type myofibers. Tissue differentiation and architecture of the models were further analyzed by histology and compared to standard 2D cultures (Fig. 3b). Whole-mount immunostaining of 3D models for Myh showed multinucleated, aligned striated myofibers indicating sarcomeric organization already at day 6 of differentiation where 2D cultures showed less densely packed and aligned fibers with only weak Myh-staining. At day 10 of differentiation, 2D cultures remained as loosely packed myofibers, albeit with stronger Myh-expression, whereas 3D models showed fully compacted and aligned myofibers. Furthermore, immunostaining for muscle-specific actin-anchoring α-actinin and staining for filamentous actin (F-actin) confirmed the myofibers alignment and striation in 3D models. Cross-sectional immunohistochemical analysis of Myh expression in the middle part of the dumbbell-shaped models showed a uniform distribution of myofibers over the entire oval-shaped area, which was typically in the order of about 1–1.5 mm width and about 200–300 μm height. Starting after about one week of differentiation, models lifted up about 1 mm from the agarose substrate and showed occasional spontaneous contractions highlighted by post bending. In summary, these data demonstrate differentiation of 3D bioprinted Matrigel/skeletal muscle cell models into structural and functional myofibers with contractile properties.

**Functionality of the printed models.** Having observed spontaneous contractility of 3D human skeletal muscle models, we developed an EPS system for controlled induction of contractions of the differentiated models in 24-well plates. The stimulation system allowed reliable controlled single twitch contractility (Supplementary Movie 1). Contractions were blocked by addition

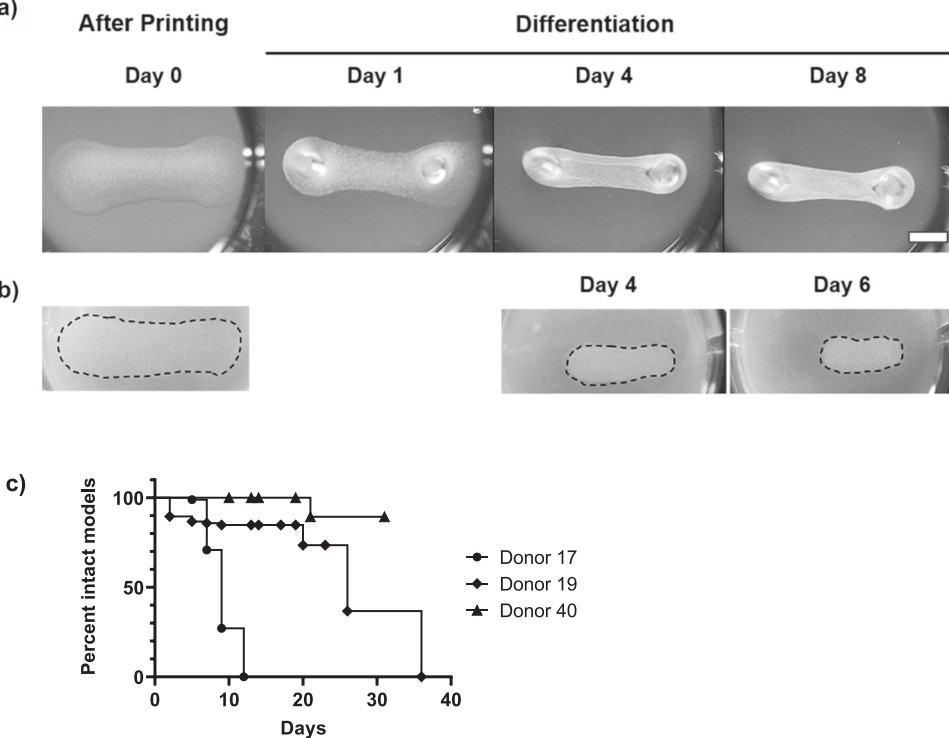

**Fig. 2 Morphological tissue development in 3D bioprinted Matrigel/cell models. a** Macroscopic images of a typical Matrigel/cell model after printing (day 0) and following fixation with two micro-posts in differentiation medium at day 1, 4 and 8. Scale bar, 2 mm. **b** Macroscopic images of a typical printed model overlaid with Matrigel after printing and during differentiation without micro-posts at day 4 and 6. **c** Time course of the survival of models from different donors until they tore between the micro-posts. Shown are the results ≥ 2 independent experiments.

of the myosin inhibitor blebbistatin (Supplementary Movie 2) indicating physiological stimulation and contractility of the 3D skeletal myofiber models. Exercising skeletal muscle stimulates anabolic pathways such as the AKT pathway[25] and the release of myokines mediating the pleiotropic systemic beneficial effects of exercise[26–28]. Thus, to further analyze the physiological relevance of our in vitro human 3D skeletal muscle models, we subjected them to a 3 h exercise protocol of repeated contractions and measured afterwards AKT pathway activation and myokine induction (Fig. 4). Akt phosphorylation was not consistently regulated in 12-day old models but was significantly induced about 6-fold in 16-day old models compared to non-exercised control models independently from the phosphorylated amino acid analyzed (Fig. 4a). The well-known myokine gene *IL-6*[29] was significantly induced in 12-day and 16-day old models, and in 19-day old models, where *IL-6* gene expression was induced 5.7-fold (Fig. 4b). These data indicate that the 3D models are valid in vitro exercise models for skeletal muscle hypertrophy and myokine stimulation.

Next, we analyzed absolute contractile force production of the 3D models. To this end, 3D models were mounted onto two hooks of an organ bath force transducer apparatus for measuring isometric contractile force upon EPS as performed for isolated mouse muscles in vitro[30]. First, we established supramaximal stimulation conditions by increasing current strength of 2 ms bipolar rectangular single twitch EPS (Fig. 5a). Peak force production increased with stimulation strength and appeared to reach a plateau of about 120 μN at 300 mA. Thus, a supramaximal stimulation intensity of 400 mA was chosen in order to be sure that all myofibers of the models were stimulated in subsequent measurements. Skeletal muscles have an optimal length for isometric twitch force when they are neither slack nor taut[30]. 3D models were mounted likewise, but great care was

required during elongation not to rupture them, because they were much less sturdy as mouse muscles. Small elongations of 20 and 40 μm of the tightly mounted models of about 8 mm length were tolerated but did not change contractile force (Fig. 5b). In contrast, a marked 3-fold force increase was observed by passing from single pulse EPS to multiple pulse EPS. Next, we investigated the correlation between peak force and EPS pulse length. Decreasing pulse length from 2 ms to 1 ms and 0.5 ms slightly reduced peak force by 9 and 21% respectively (Fig. 5c). The length of the electric pulse determines the site of excitation of isolated skeletal muscle[31]. Particularly, pulses longer than 1 ms directly activate sarcoplasmic $Ca^{2+}$-release without involving normal muscle action potentials via voltage-dependent $Na^+$-channels. Thus, to test whether the contractions induced by 1 ms EPS were mediated by physiological $Na^+$-dependent action potentials, models were exposed to 1 μM TTX, which fully inhibits action potential generation and T-tubular $Na^+$-currents[32]. TTX completely suppressed contractions induced with 1 ms EPS (Fig. 5d). Overall, these data suggest that the 3D bioprinted myofiber model is a valid human microphysiological in vitro assay for the screening and study of skeletal muscle exercise contractility and myokine production.

**Contractile force increases by known skeletal muscle stimulators.** To corroborate the physiological relevance of our contractility model, we studied the effects of known stimulators of skeletal muscle force. First, we tested caffeine. It is the most widely used physical performance enhancer worldwide[33]. However, its mode of action is still debated as supraphysiological concentrations are required to stimulate skeletal muscle contractility in vitro[34]. Nevertheless, we observed similar increases in EPS-induced contractile peak force (+43.4%) by high concentrations of 10 mM

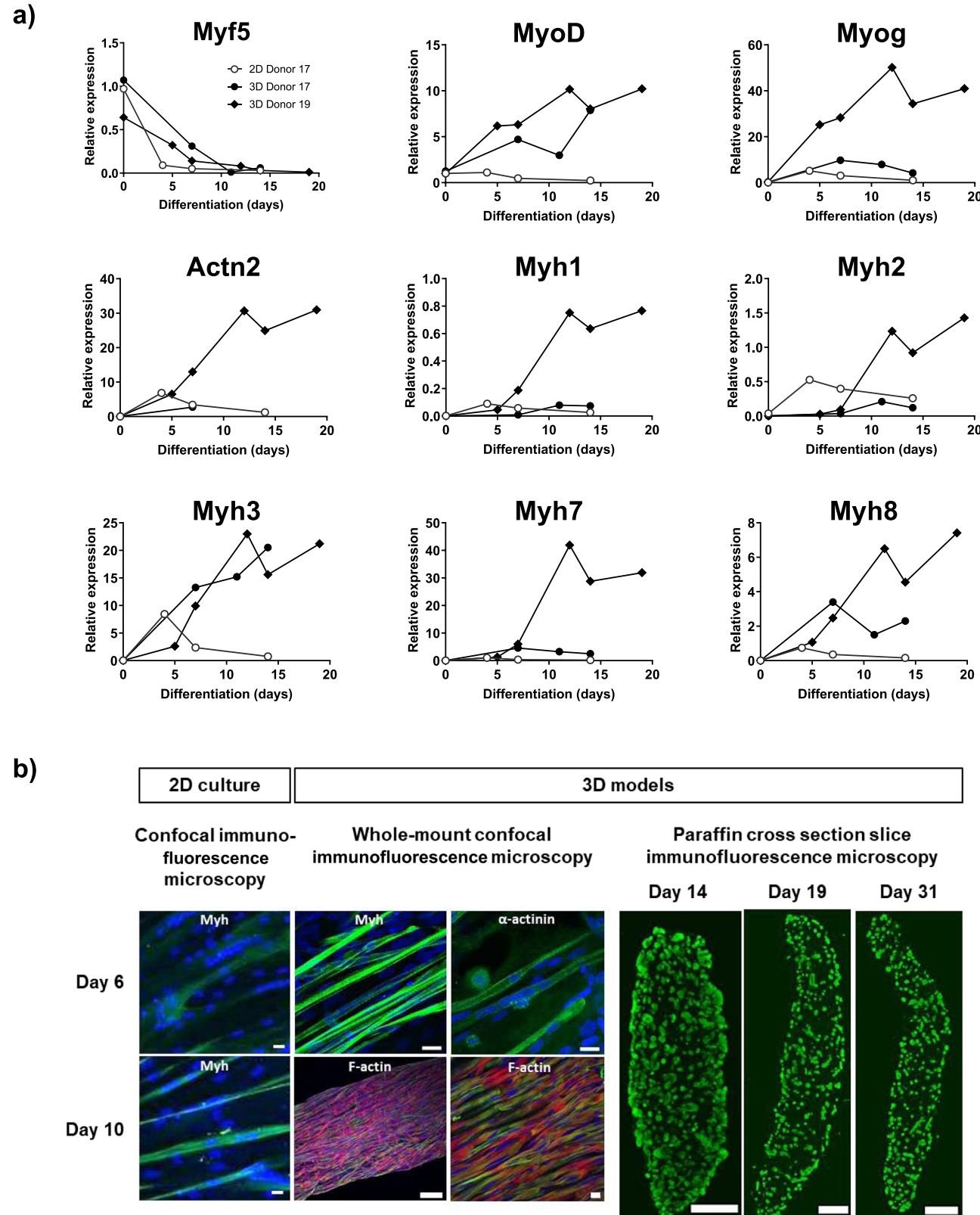

caffeine (Fig. 6a). Furthermore, caffeine increased much more the duration of the EPS-triggered contraction (+413% width50 and +693% area under curve [AUC]). Next, we tested the skeletal muscle troponin C activator Tirasemtiv or formerly called CK-2017357[35]. It acutely improves muscle force by binding to the fast-skeletal-troponin complex, which slows calcium release. Addition of 20 µM Tirasemtiv enhanced EPS-induced contractile force within a few minutes to a sustained level for up to 30 min (Fig. 6b). In contrast, solvent control (0.2% DMSO) did not lead to significant force changes (Fig. 6d). Tirasemtiv significantly increased peak force

**Fig. 3 Differentiation of 3D bioprinted Matrigel/skeletal muscle cell models. a** Marker gene expression analyses. Temporal expression profiles of myogenesis marker genes (*Myf5, MyoD, Myog, Actn2, Myh1-3, Myh7&8*) in differentiating 2D cultures of skeletal muscle precursor cells from a 17-year-old donor and in 3D bioprinted models with cells from a 17-year-old and a 19-year-old donor. Gene expression was determined by qPCR and normalized by *18 S, GAPDH, TBP* and *β2M* housekeeping genes. Expressions were standardized by arbitrarily taking *MyoD* expression at day 0 as 1. Shown are the means of ≥ 2 independent experiments. **b** Histological analyses of tissue cultures of a 17-year-old donor. Confocal immunofluorescence microscopy of Myosin heavy chain (Myh) expression in control 2D cultures of differentiation day 6 and 10 as indicated in the figure. Whole-mount confocal microscopy images of models differentiated for 6 and 10 days were immunostained for Myh, α-actinin and F-actin proteins as indicated in the figure. Nuclei were stained with DAPI. Scale bar, 20 μm, except for F-actin, 200 μm (left image). Cross section microscopy images of models from a 40-year-old donor after 14, 19 and 31 days of differentiation were immunostained for Myh. Scale bar, 150 μm.

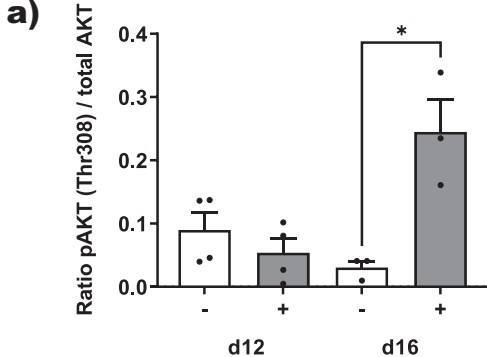

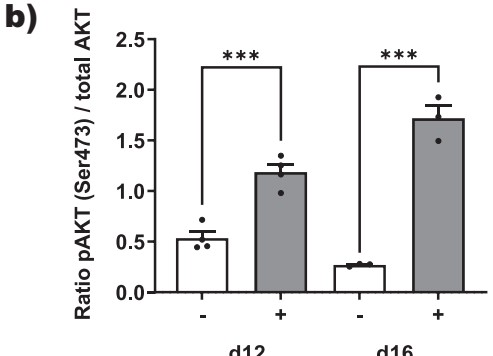

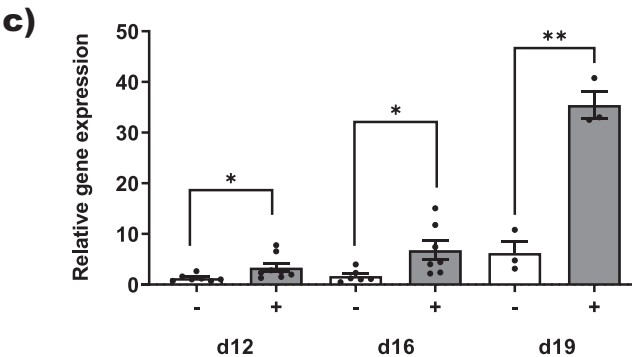

**Fig. 4 EPS-induced repetitive model contractions activate Akt hypertrophy pathway and induce IL-6 myokine expression.** 3D models (*n* ≥ 3) from a 19-year-old donor, differentiated for 12 (d12), 16 (d16) and 19 days (d19), were stimulated by high impact EPS (+) for 3 h or left untreated as controls (−). Induction of Akt phosphorylation as a ratio of total Akt was determined by Western Blotting. **a** Thr308 phosphorylation. **b** Ser473 phosphorylation. **c** Relative induction of *IL-6* gene expression determined by qPCR and normalized by *GAPDH* expression. Shown are means ± sem. Statistics: Unpaired *t*-test, $*p < 0.05$, $**p < 0.01$, $***p < 0.001$.

by 46.3%, width50 by 60.8% and AUC by 148% (Fig. 6c), whereas solvent control (0.2% DMSO) showed only a small peak force decrease of 9.2% and no significant changes of width50 and AUC (Fig. 6e). A limited dose-response curve of Tirasemtiv (10–30 μM) indicated a half-maximal effect at 10 μM and a maximal effect at 20 μM (Fig. 6f). In summary, two known stimulators of skeletal muscle strength (caffeine and Tirasemtiv) increased similarly contractile forces in our 3D human myofiber models.

## Discussion

In the present study, we report the generation of functional human skeletal muscle models in 24-well plates using microvalve-based DOD printing of cooled Matrigel/precursor cell suspensions in an automated 3D bioprinting platform. To our knowledge, this is the first description of high precision DOD 3D bioprinting of human tissue models using pure Matrigel solutions as a hydrogel bioink loaded with cells. Previously, extrusion-based 3D bioprinting of cell-laden Matrigel has been reported[36]. However, Matrigel printing difficulty due to thermal gelling at RT allowed only single layer printing, despite an intricate temperature-controlled enclosure system for the 3D printer[22] or required mixing Matrigel with alginate or agarose to allow proper rheological properties for printing, which, however, greatly impaired tissue differentiation[20,36]. Horvath and colleagues printed thin layers of Matrigel without cells as basement membranes for subsequently printing of cells using microvalve-based 3D bioprinting in contact extrusion mode[23].

Our cooling system provided easy and reliable refrigeration of the printing cartridges such that gelling of the Matrigel/cell suspension was constantly avoided. A cooling temperature of 7 °C was chosen as the optimal temperature, because it reliably prevented Matrigel gelling during printing and did not yield excessive water condensation from the air at the outside of the cooling jacket, which would have dripped into the well plate during printing. We did not experience any gelling issues in the microvalves, although the measured temperature at the orifice being 13 °C was clearly higher than the 7 °C set temperature of the cooling system. Nevertheless, inclusion of the microvalve in the cooling jacket seems like a reasonable future improvement of the cooling system to further stabilize temperature control of the whole printhead and thereby enhancing printing reliability. Our Matrigel gelling tests identified that total protein concentrations of 8–10 mg/mL were best for printing and subsequent solidification at 37 °C independently of the two batches tested. Furthermore, these high density extracts apparently showed negligible cell sedimentation for up to 1.5 h such that no cartridge stirring system was required during printing. Although we did not observe prominent Matrigel batch differences regarding rheological and gelling properties during printing, when using the same total protein concentration, we recommend testing different batches of Matrigel for optimal printability and tissue differentiation. Drop formation and clogging at the microvalve nozzle was a serious issue for re-usage of microvalves, before we established a more rigorous cleaning

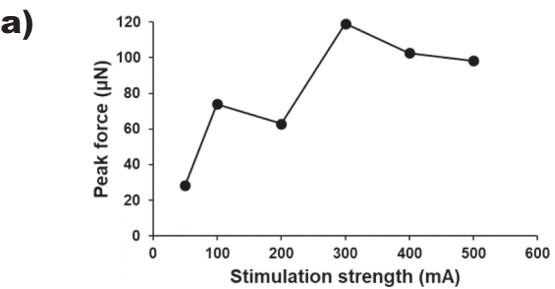

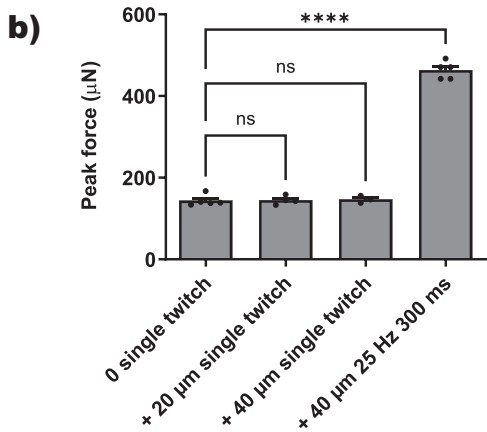

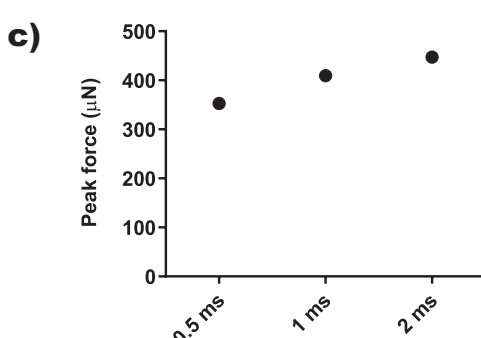

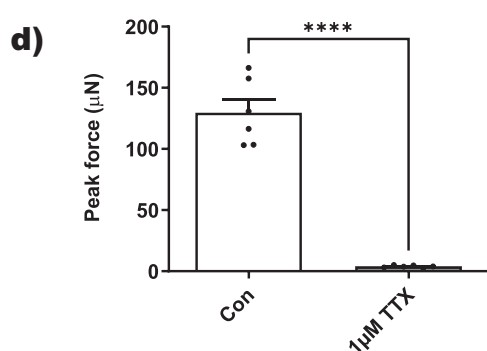

**Fig. 5 Measurement of EPS-induced contractile force. a** Representative 2 ms single twitch EPS-induced contractile peak forces of a 3D model from a 19-year-old donor, differentiated for 18 days, in relation to pulse strength. **b** Effect of model elongation and single versus multiple twitch activation on 400 mA EPS-induced contractile peak force production in a representative model ($n \geq 3$ stimulations) Shown are means ± sem. Statistics: One-way ANOVA, Bonferroni's Multiple Comparison test, ****$p < 0.0001$. **c** Effect of EPS (50 Hz, 300 ms, 400 mA) pulse length on contractile peak force in a representative model. **d** Effect of 1 μM TTX or solvent control (Con) on EPS-induced (1 ms pulse, 25 Hz, 300 ms, 400 mA) contractile peak force in a representative model from a 19-year-old donor differentiated for 17 days ($n = 6$ stimulations). Shown are means ± sem. Statistics: Unpaired $t$ test, ****$p < 0.0001$.

same well plate 3D bioprinting platform as used in this study[14]. However, tissue differentiation was very poor, because the utilized synthetic bioink was non-permissive for cell growth and differentiation, and the posts of the cell culture inserts were too stiff to allow contractions of developing myofibers. Vandenburgh et al.[11] used flexible posts made of polydimethylsiloxane (PDMS) to generate differentiated muscle tissues. PDMS is known for its ease of use and processing and is thus frequently employed as a biocompatible polymer in micro-fluidic device development. Unfortunately, the material exhibits strong absorption of small hydrophobic molecules and makes it unsuitable in combination with drug testing[37]. Now, we could solve both issues by using cell-laden pure Matrigel as bioink in combination with very flexible micro-posts stuck in an agarose bed. We were using the same cell density as previously reported by Laternser and colleagues[14]. Since muscle tissue is mainly comprised of cells, and has little extracellular matrix (ECM), we used the highest possible cell density, which doesn't lead to clogging of the microvalve.

This allowed the differentiation of contracting 3D myofiber tissue models that consisted of aligned, striated and uniformly 3D dispersed myofibers as shown by immunohistochemistry. Furthermore, we demonstrated the necessity of attachment points for myofiber formation and differentiation in comparison to non-attached models, which condensed in all directions but did not show any morphological signs of myofiber development or spontaneous contractions. These data strongly suggest that 3D skeletal muscle cell models do not self-organize into aligned contractile myofiber tissue without attachment points. Consistently, the critical importance of structural and mechanical support for 3D skeletal muscle tissue engineering is well documented[38]. In addition, our experiments strongly indicate that optimal differentiation of skeletal muscle tissue models requires flexible attachment points with matched resistance for the contractile forces of the developing myofibers. This is probably similar as during dynamic development of the musculoskeletal system in the embryo where contractile forces by the developing muscle induce critical mechanical signals for proper joint organogenesis with evolving and adapted connections between bones, tendons, ligaments and muscles[39]. Models made of cells from a 19-year-old donor differentiated better, based on marker gene analysis, and ruptured much later than models from a 17-year-old donor. This suggests that model rupture is not due to strongly matured myofibers that create too high contractile forces for tissue cohesion. Instead, model rupture may be caused by general tissue condensation as seen in models without posts, when there are no or not enough functional myofibers that hold the tissue together. Nevertheless, according to the *Myh* expression pattern the in vitro engineered 3D models from both donors mainly consisted of slow and embryonal type myofibers. In any case, cell culture inserts with micro-posts of defined variable

procedure (Supplementary Fig. 1). Initially, microvalves were cleaned after printing by three washing cycles with 70% ethanol according to bioprinter's basic procedure. However, this resulted often in impurities and precipitations in the microvalves.

Recently, we generated human skeletal muscle models on postholder cell culture inserts consisting of alternating layers of photo-polymerized GelMA-PEGDA bioink and cells using the

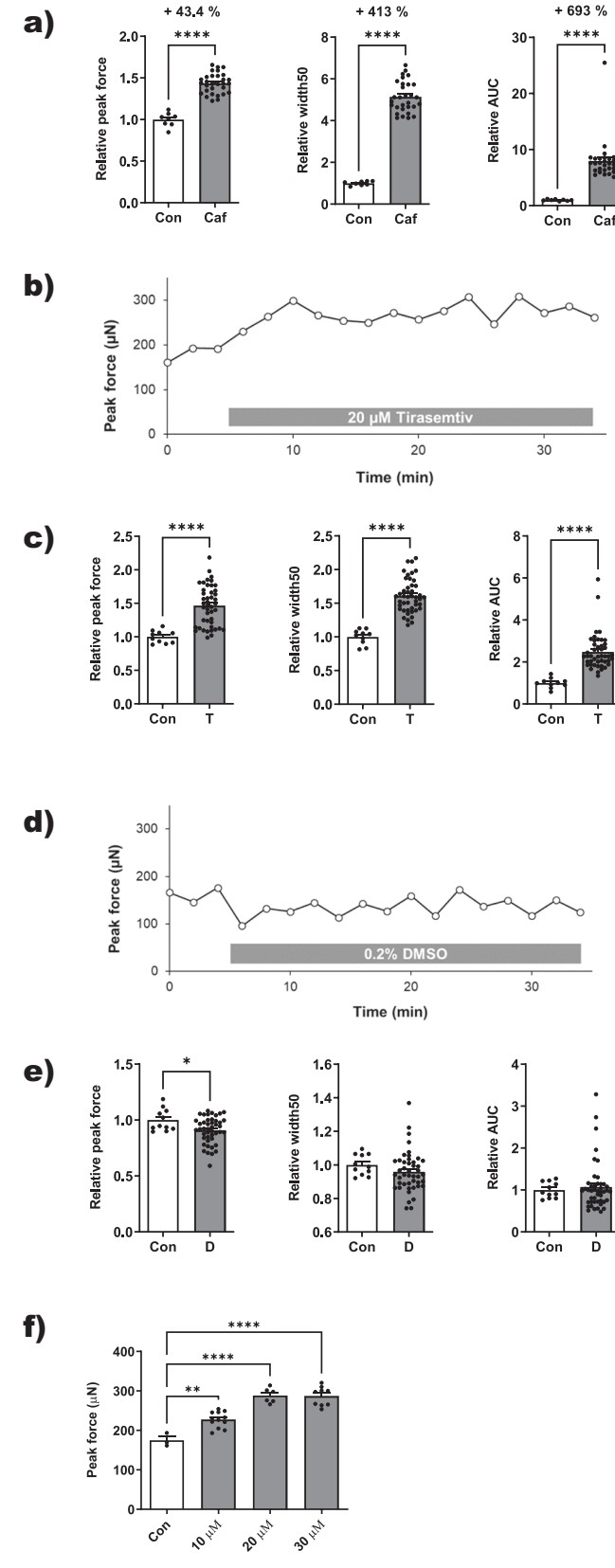

**Fig. 6 Acute enhancement of EPS-induced contractile force by known muscle stimulating drugs. a** Effect of 10 mM caffeine (Caf) or solvent control (Con) on EPS-induced peak force and contraction duration (width50 and AUC [area under curve]) in 3D models from a 19-year-old donor differentiated for 20 days ($n \geq 8$). Data are presented as means ± sem. Statistics: Unpaired *t*-test; ****$p < 0.0001$. **b** Time course of the effect of 20 μM troponin C activator Tirasemtiv on peak force production in a representative model from a 19-year-old donor differentiated for 18 days. **c** Relative effect of 20 μM Tirasemtiv (T) on peak force and contraction duration compared to before treatment (Con). Shown are means ± sem ($n \geq 10$). Statistics: Unpaired *t*-test, ****$p < 0.0001$. **d** Time course of the effect of 0.2% DMSO solvent on peak force production in a representative model from a 19-year-old donor differentiated for 20 days. **e** Relative effect of 0.2% DMSO (D) on peak force and contraction duration compared to before treatment (Con). Shown are means ± sem ($n \geq 11$). Statistics: Unpaired *t*-test, *$p < 0.05$. **f** Dose-response relation of the effect of Tirasemtiv on EPS-induced peak force in a representative model from a 19-year-old donor differentiated for 29 days. Shown are means ± sem ($n \geq 3$). Statistics: One-way ANOVA, Dunnett's multiple comparison test, **$p < 0.01$, ****$p < 0.0001$.

of skeletal muscle ECM. In vivo, poly-nucleated myofibers are densely packed in bundles and only coated by a thin basement membrane as ECM. Matrigel is a solubilized basement membrane extract of a mouse sarcoma rich in the common major components of basement membranes including laminin, collagen IV, heparin sulfate proteoglycans and nidogen. Thus, it is ideally suited as physiological bioink for in vitro skeletal muscle tissue engineering. To our knowledge, we are the first to use pure Matrigel as scaffolding hydrogel bioink for 3D bioprinting of skeletal muscle tissue. Matrigel is still the gold standard of support matrix for in vitro engineering of 3D human tissues and organoids[19]. Therefore, our Matrigel-based 3D bioprinting method may be used for many other in vitro 3D anisotropic tissue models that do not self-organize from precursor cell spheroids but require complex topological arrangements of cells and/or different cell types. In this respect, we see a great potential of cell-laden Matrigel 3D bioprinting for in vitro tissue engineering of more predictive complex human MPS for drug discovery and development.

Our custom-made EPS system for 24-well cell culture plates (Supplementary Fig. 2) is the first electrical stimulation system allowing well-by-well or column-by-column stimulation in a standard well plate format providing medium throughput assay capacity. We have chosen an electrical excitation of two U-shaped platinum (Pt) electrodes per well that are parallel to the long axis of the muscle model. This configuration was favored in the seminal study of Cairns and colleagues on the nature and method of proper electrical stimulation of isolated skeletal muscles in vitro. It provides a uniform electrical field along the aligned myofibers triggering action potential simultaneously[31]. To date, most in vitro studies of electrically stimulated cultured muscle cells have used commercial or homemade systems consisting of parallel carbon plate electrodes[43]. As carbon electrodes are difficult to clean and maintain for reuse, because they absorb salts, electrolysis by-products and proteins from the media and cultured cells, we selected metabolically and chemically inert Pt electrodes. They are easily cleaned and sterilized by 70% ethanol. In addition to electrode material and configuration, electrical pulse characteristics and patterns are decisive for the extent of skeletal muscle excitation[31]. First, we determined minimal pulse strength of 10 V and 400 mA for reaching maximal contractility of the models in the in vitro exercise assay and force measurement assay, respectively. Similarly, as shown for isolated mouse

stiffness and models from many more donors at different age will be required to analyze the relationships between post deflection with respect to stiffness, tissue differentiation, model rupture and different donor age. To date, most in vitro engineered skeletal muscle tissue models are made of fibrin-containing Matrigel hydrogels[12,40–42], even though fibrin is not a natural component

flexor digitorum brevis (FDB) muscle fibers[44], this is very likely the activation strength where all myofibers, having different thresholds stimulus strength, are activated. Secondly, it is important to select a pulse length which yields maximal contractile force by inducing action potentials at the sarcolemma-like physiological neuromuscular excitation, but which does not bypass voltage-dependent membrane $Na^+$-channels by directly triggering $Ca^{2+}$- release from the sarcoplasmatic reticulum[31]. We have chosen 1 ms pulses and shown that TTX completely blocked contractions demonstrating neuromuscular-like membrane action potential excitation. Furthermore, we have chosen a protocol of 300 ms 25 Hz pulse trains every second for 3 h as a short-term high impact in vitro exercise model and validated it by showing *IL-6* myokine gene expression induction and activation of Akt hypertrophy pathway. Both are well established short-term high impact exercise markers in vitro as well as in vivo[42,43,45]. Thus, our human 3D skeletal myofiber model is a new promising functional model to analyze the effect of exercise on muscle hypertrophy and myokine production in vitro. Finally, we showed that our 3D models acutely respond to the known human muscle force inducers caffeine and Tirasemtiv. Within about 5 min, 10 mM caffeine significantly increased EPS-induced peak force by almost 50% and markedly prolonged contraction. A similar acute peak force increase, but not contraction prolongation, by 5 mM caffeine was recently reported for mouse FDB muscle single fibers using similar EPS (1 ms pulse, 25 Hz, 350 ms duration) conditions[34]. Contraction prolongation by caffeine in our 3D model, but not in FDB muscle fibers may indicate that the myofibers in our model are immature and that excitation-contraction coupling is not yet fully developed as in the ex vivo mature FDB fibers. This is also supported by our myofiber differentiation assessment based on marker gene expression analyses. The troponin C activator Tirasemtiv (CK-2017357) is an experimental drug for treating neuromuscular diseases. It increases muscle force by augmenting the retention time of calcium at the troponin C complex. Infusion of Tirasemtiv enhanced peak force by about 40% within minutes in an in-situ rat extensor digitorum longus muscle preparation[35]. Our models showed a very similar rapid highly significant force increase (+46%) by 20 μM Tirasemtiv. At this concentration, it was shown to be a specific full agonist of fast skeletal myofibrils with neglectable activation of slow and cardiac myofibrils[35]. Since our gene expression fiber typing assessment indicated mostly embryonal, postnatal and slow fibers, we hypothesize that Tirasemtiv is also a high affinity activator of embryonal and/or postnatal fibers.

To our knowledge, our 3D bioprinted human skeletal muscle model is the first in vitro MPS that reproducibly, robustly and highly significantly mimics exercise and force increases by known human muscle stimulating drugs. In addition, it allows medium throughput drug screening. Thus, it is a highly promising new functional 3D MPS for the identification and development of novel drugs for the treatment of muscle wasting diseases.

## Methods

**Matrigel gelling tests**. Two different batches of frozen, phenol red free, standard Matrigel matrix (Corning, cat. no. 356237; protein concentration 8–12 mg/mL) were thawed at 4 °C overnight. Dilutions were prepared on ice with pre-chilled MyoTonic serum-free growth medium (COOK MyoSite, cat. no. MK-2288). Droplets of 50 μL ($n = 3$) were pipetted onto microscope glass slides on ice. A second cooled glass slide with 1 mm spacers was laid on top. After incubation at 37 °C for 30 min, the top glass slide was removed and the form stability of the Matrigel droplets was qualitatively assessed by optical inspection of the rheological properties during lifting-up with a spatula.

**Cell culture and 3D bioprinting**. Primary human skeletal muscle-derived cells (hSkMDC) were bought from different donors (COOK MyoSite, cat. no. SK-1111; lot P101082-17M5, 17-year-old donor; P201076-19M, 19-year-old donor and P201028-40M, 40-year-old donor) and cultured as previously reported[14]. Briefly,

cells were thawed according to the manufacturer's recommendation and cultured for 4-5 days in MyoTonic serum-free growth medium containing 20% fetal bovine serum (FBS; ThermoFisher, cat. no. 16000), 10 μg/mL insulin (AMIMED, cat. no. 5-79F00-G) and 50 μg/mL gentamycin (ThermoFisher, cat. no. 15750) until they reached about 60% confluence. Cells were harvested with Accutase (Sigma, cat. no. A6964) after washing twice with DPBS (ThermoFisher, cat. no. 14190136). Following centrifugation, the cell pellet was resuspended on ice by gentle pipetting in pure Matrigel (protein concentration 8–10 mg/mL) at a concentration of $2 \times 10^7$ cells/mL. The cell-laden Matrigel suspension was loaded into printing cartridges pre-cooled at 7 °C, which were mounted on the cooled printhead of a 3DDiscovery 3D bioprinter (regenHU). The water-based printhead cooling system was controlled by a water bath Pilot One (Huber) with an integrated external temperature sensor at the cooling jacket.

Dumbbell-shaped 4-layer cell/Matrigel models were 3D bioprinted by DOD mode using microvalves with 150 μm orifice (regenHU, cat. no. CF300) on a 3D microprinter. Microvalve printing parameters were 100 μs stroke, 200 μs opening time, 0.04 mm dosing distance and 600 hPa pressure. Models were printed in 24-well plates (Weidmann Medical Technology AG) at RT on a 0.8% agarose substrate (Sigma, cat. no. A9918). Agarose substrate (950 μL/well) was produced by dissolving agarose in growth medium without supplements. Every 30 min, the printing process was stopped, and 5 single droplets were printed on glass slides. The diameter of each droplet was measured using a microscope (Zeiss, Axiovert 200 M) and the corresponding software (Zeiss, Zen). After printing, 3D dumbbell-shaped models were incubated for 1 h in a 5% $CO_2$ incubator at 37 °C to achieve full solidification of the Matrigel. Subsequently, two tube posts of about 1 cm length and 0.9 mm diameter cut near the smaller end of pipette tips (Mettler Toledo Tips GelWell, cat. no. GT-250-6) were manually inserted in the center of the two extremities of the printed dumbbell-shaped models, before adding growth medium and incubating at 37 °C in a 5% $CO_2$ incubator. After printing, microvalves were cleaned using a microvalve cleaning kit according to the manufacturer's recommendations (Fritz Gyger, cat. no. 21986). Valves were first washed 5-times backwards with sterile hot water (70 °C) followed by 3-times washing with 70% ethanol and 3-times air drying using the syringes of the kit.

**Tissue differentiation**. After printing and tube post insertion, 3D models were incubated for 24 h in growth medium at 37 °C with 5% $CO_2$. Differentiation was induced by replacing growth medium with differentiation medium, which consisted of phenol red free Dulbecco's modified Eagle's medium (DMEM) high glucose (GIBCO, cat. no. 31053) containing 2% heat inactivated horse serum, hiHS, (GIBCO, cat. no. 26050), 1% FBS (GIBCO, cat. no. 16000), 1% chicken embryo extract (MP Biomedicals, cat. no. 2850145), 1% sodium pyruvate (GIBCO, cat. no. 11360), 1% L-glutamine (GIBCO, cat. no. 25030) and 50 μg/mL gentamycin (GIBCO, cat. no. 15750) by incubation at 37 °C with 7.5% $CO_2$. Differentiation medium was changed every 2-3 days. Muscle cells in 2D culture were differentiated similarly by incubation in DMEM high glucose with pyruvate (GIBCO, cat. no. 31966) containing 2% hiHS, 1% FBS 1% chicken embryo extract and 50 μg/mL gentamycin.

**Marker gene analysis by qPCR**. Total RNA was isolated from models using RNeasy Mini Kit (Qiagen, cat. no. 74104). Gene expression analyses were performed by qPCR using High Capacity cDNA reverse transcription kit (Thermo-Fisher, cat. no. 4374966), TaqMan gene expression master mix (ThermoFisher, cat. no. 4369016) and corresponding TaqMan assays (Applied Biosystems, cat. nos. listed in Supplementary Table 1) according to the manufacturer's recommendation. The housekeeping genes *18 S*, *GAPDH*, *TBP* and *β2M* were used for gene expression normalization.

**Histological analyses**. 3D models were washed twice in PBS and then fixed in 4% paraformaldehyde (ProSciTech, cat. no. EMS15714-S) in PBS at 4 °C for 30 min. For paraffin section histological analyses, fixed models were washed with PBS and embedded in 2% agarose. After dehydration using a tissue processor (Leica, cat. no. ASP200), models were embedded in paraffin (Leica, cat. no. 39603002). Transverse sections (5 μm) were cut through the model with 20 μm intermediate slices using a microtome (Zeiss). Sections were dried for more than 6 days at RT. Then, they were deparaffinized followed by heat-induced (98 °C) epitope retrieval in 1 mM EDTA (Fluka, cat. no. 03679) pH 8.0 for 20 min. MYH immunostaining was performed using monoclonal mouse IgG anti human MyHC clone A4.1025 primary antibody (Upstate, cat. no. 05–716) 1:1000 in TBS buffer (Cell Signaling, cat. no. 12498 S) containing 5% BSA (Sigma, cat. no. A8806). As a control, mouse IgG isotype control (BIO RAD, cat. no. MCA 1209) was used with the same dilution. After PBS washings, goat anti-mouse IgG, IgM, IgA (H + L), Alexa Fluor™488 (Life Technologies, cat. no. A10667) was used 1:200 in TBS buffer containing 5% BSA as secondary antibody for 30 min. Nuclei were counterstained with 0.5 μg/mL DAPI (PromoCell, cat. no. PK-CA707-40043) in TBS buffer. Finally, slides were mounted with Mowiol (Sigma, cat. no. 81381) and imaged by confocal microscopy (Olympus, Fluoview FV3000).

For whole mount fluorescent immunohistochemistry, tissue clarification was performed as described[46]. Briefly, fixed models were rinsed in PBS and incubated at 37 °C for 5 days in reagent 1 consisting of 25 wt% urea (Sigma, cat. no. U5378),

25 wt% N,N,N',N'-tetrakis(2-hydroxypropyl)ethylenediamine (Sigma, cat. no. 122262), and 15 wt% polyethylene glycol mono-pisooctylphenylether (Nacalai Tesque, cat. no. 25987-85)/Triton X-100 (Sigma, cat. no. X100). Then, models were washed twice in PBS for 30 min and incubated with primary antibody or probe diluted in 0.1% Triton X-100/PBS/0.5% BSA (Sigma, cat. no. A7030)/0.01% Sodium Azide (Fluka, cat. no. 71289) for 1 day at 37 °C. In case of incubation with a secondary antibody, samples were washed in 0.1% Triton X-100 in PBS and incubated with secondary antibody for 1 day at 37 °C. 1 µg/mL DAPI (Roche, cat. no. 10236276001) was optionally added to reagent 1 solution or to the primary/secondary antibody solution for a nuclear counterstaining. After washing in 0.1% Triton X-100 in PBS, models were immersed for 24 h at RT in 20% sucrose (Sigma, cat. no. 84100) and then in reagent 2 at 37 °C with gentle shaking until complete clearing. Reagent 2 consisted of 50 wt% sucrose, 25 wt% urea, 10 wt% 2,2',2"-nitrilotriethanol (Wako Pure Chemical Industries, cat. no. 145-05605). Primary antibodies were monoclonal mouse IgG to human Myosin Heavy Chain, clone A4.1025 (Upstate, cat. no. 05–716) and monoclonal mouse IgG1 to α-actinin both diluted at 1/1000 (Sigma, cat. no. A7811). Alexa Fluor™594-Phalloïdin (ThermoFisher, cat. no. A12381) was used at 1/40 dilution for F-actin detection. Samples were washed twice for 1 h in 0.1% Triton X-100/PBS prior to incubation with secondary antibody Alexa Fluor™488-goat anti-mouse IgG (ThermoFisher, cat. no. A10667) used at 1/200. Images were acquired with a laser scanning confocal microscope (Zeiss, LSM710) equipped with a 40x oil objective and appropriate filters for Alexa Fluor™488 and Alexa Fluor™594 detection.

**Contractility analyses.** Whole model contractility and postholder movements in 24-well plates were observed and video recorded using a M80 stereo microscope with integrated IC90 E camera and Application Suite software (Leica). To induce controlled model contractions, we built a custom-made EPS system for 24-well plates. The system consisted of an electrical circuit board as well plate lid containing two U-shaped Pt electrodes in parallel to the model for each well (Supplementary Figure 2). The board was connected via a ribbon cable to a PowerLab 8/35 data control and acquisition system (ADinstruments). An electrical stimulator (Hugo Sachs Elektronik, Stimulator C type 224) delivered bipolar rectangular pulses. The muscle myosin inhibitor blebbistatin (Sigma, cat. no. B0560) was used at 10 µg/mL final concentration and the voltage-gated sodium channel TTX (Tocris, cat. no. 1078) at 1 µM final concentration. High impact exercise-mimicking induction of model contractions was performed by repeated EPS for 3 h using trains of 25 Hz, 10 V, 1 ms pulses for 300 ms every second. To analyze Akt phosphorylation, models were washed with ice-cold PBS immediately after EPS and were lysed in 300 µL RIPA buffer (Sigma, cat. no. R0278) containing Halt protease and phosphatase inhibitor cocktail (ThermoFisher, cat. no. 78440) for 5 min on ice followed by freezing at −80 °C. Total protein content of the lysates was determined using BCA protein assay kit (ThermoFisher, cat. no. 23227). Phospho- and total Akt- protein levels were analyzed by SDS-PAGE/Western blotting. 10 µg total protein in a total of 30 µL Laemmli buffer (BioRad, cat. no. 161–0747) were boiled for 5 min at 95 °C and loaded onto a NuPage 4 to 12% Bis-Tris mini protein gel (ThermoFisher, cat. no. NP0336BOX) that was run for 10 min at 100 V followed by 45 min at 200 V in NuPage MES SDS running buffer (ThermoFisher, cat. no. NP0002). Electrophoresed proteins were transferred onto nitrocellulose membrane (BioRad, cat. no. 1704158) using a Trans-Blot Turbo transfer system (BioRad, cat. no. 1704150EDU) for 7 min at 2.5 A with a gradient up to 25 V. Non-specific protein binding was blocked by incubating the membrane in TBS containing 0.1% Tween 20 (Merck, cat. no. P9416) and 5% skim milk (Merck, cat. no. 70166) for 1 h at RT under gentle agitation. Phosphorylated Akt species were detected by incubating membranes overnight at 4 °C with Phospho-Akt (Thr308) (D25E6) XP rabbit monoclonal antibody (Cell Signaling, cat. no. 13038) diluted at 1/1000 in TBS containing 0.1% Tween 20 (TBST) and 5% BSA (Sigma, cat. no. A4503) and by Phospho-Akt (Ser473) (D9E) XP rabbit monoclonal antibody (Cell Signaling, cat. no. 4060) diluted at 1/2000 in the same buffer. Total Akt was detected by rabbit Akt antibody (Cell Signaling, cat. no. 9272) diluted at 1/1000 using the same incubation procedure. After the incubation with primary antibody, membranes were rinsed with TBST followed by three washes with TBST for 5 min at RT on a shaker. Then, membranes were incubated with anti-rabbit IgG, HRP-linked antibody (Cell Signaling, cat. no. 7074) secondary antibody diluted at 1/2000 in TBST containing 5% skim milk for 1 h at RT under shaking. Membranes were rinsed with TBST followed by three washes with TBST for 5 min at RT on a shaker. Finally, HRP activity was detected by chemiluminescence using ECL Prime Western Blotting system (GE Healthcare, cat. no. RPN2232). Luminescence was measured using a Fusion FX7 Spectra multispectral imaging system (Witec AG).

**Force measurements.** To measure EPS-induced absolute contractile force of models, we used an ex vivo organ-bath contractility assay for isolated mouse muscles similarly as described[30]. Models were mounted on two hooks that were connected to a force transducer (WPI, cat. no. SI-KG2) and a micromanipulator in a horizontal tissue bath (WPI, cat. no. SI-HTB2) using the model's two holes left from post removal. A current-controlled electrical stimulator (Ugo Basile, cat. no. 3165) delivered unipolar EPS excitation consisting of 400 mA, 1 ms pulses at 25 Hz for 300 ms. Force measurements were amplified by a force transducer amplifier (WPI, cat. no. SI-BAM21-LCB), controlled by a data acquisition system

(ADInstruments, cat. no. PowerLab 8/35) and analyzed by LabChart V8 software (ADInstruments). A digital oscilloscope (Fluke ScopeMeter 190–202) was used to confirm the frequency, shape and amplitude of the delivered electrical pulse signals. Caffeine was bought from Sigma (Cat. no. C8960) and Tirasemtiv[35] was synthesized at Novartis.

**Statistics and reproducibility.** Data are expressed as mean ± sem. The number of replicates and the statistical methods (GraphPad Prism) used are specified in the corresponding Figure legends.

**Reporting summary.** Further information on research design is available in the Nature Research Reporting Summary linked to this article.

## Data availability
All data generated or analyzed during this study are included in this published article (and its supplementary information files).

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

## Acknowledgements

We would like to thank Marie Ronco for expert technical help in force measurements. Furthermore, we thank Bernhard Zahnd, Simone Tavani and Matthias Brechbühl from the engineering center of NIBR, Novartis, Basel for designing, producing and assembling the 24-well EPS system. Finally, we thank regenHU for the development of a printhead cooling system.

## Author contributions

H.K. and M.Ri. designed experiments and H.K., M.Ri. and E.B. wrote the manuscript. A.A., S.A. and D.B. established and optimized Matrigel 3D bioprinting of human skeletal muscle models and performed qPCR marker gene differentiation analyses. A.A., S.A., N.A., H.J., M.S. and A.D carried out immuno-histological and microscopical examinations. M.Ra. oversaw microscopy. H.K. and M.Ra. supervised the engineering of the electrical circuit board and system for 24-well plate EPS. A.A., D.B. and S.A. investigated EPS-induced model contractility and A.A. performed EPS-induced Akt and IL-6 inductions, force measurements and pharmacological regulations. H.J. supervised the efforts, including the manuscript preparation.

## Competing interests

The authors declare no competing interests.
