## [Peer Review File · Communications Biology]

Reviewers' comments:

Reviewer #1 (Remarks to the Author):

The paper reports a droplet-on-demand bioprinting protocol of primary-derived human muscle cells in Matrigel. The developed muscle model exhibits high levels of differentiation and is capable of recapitulating many physiological functions, using various immunohistological, electrophysiological and molecular biological experiments to back up the claims.

Although the work adds a new aspect for handling pure Matrigel in a bioprinting context, specifically using temperature control and a set concentration range for Matrigel solution bioinks, the physiological constructs made using the droplet-on-demand protocol are not particularly impressive in terms of droplet spatial resolution and overall printed construct size or 3D structures, so the merit of this work in the area of biomaterials is somewhat tenuous. The paper could be improved by investigating the fundamental relations between temperature and Matrigel rheology and improve printing resolution. Furthermore, the use of tumorigenic Matrigel as the bioink in this study limits the translational utility of this bioprinting protocol as researchers seek other highly bioactive alternatives that avoid the pitfalls of Matrigel.

There are also several lapses in writing mechanics in the discussion section and elsewhere that obscure the meaning of the paper. The specific lapses are listed below:

- Page 12, a comma between "too stiff" and "not allowing" should be present.
- Page 17, the sentence beginning with "although the measured..." is unclear and should be rephrased.
- Page 17, the sentence beginning with "although, we did not..." should have the comma between "although" and "we did not..." removed.
- Page 18, the sentence beginning with "Drop formation" should have comma between "microvalve" and "before" removed.
- Page 18, the sentence beginning with "Furthermore, we demonstrated..." should replace "nor" with "or".

In addition to the text itself, the figures describing the results have significant instances of unclarity. In Figure 1c, it is not clear what length is represented by the scale bar. Is it 2mm, as indicated at the end of the Figure 1 caption, or something else? In Figure 4, although the caption indicates that there should be some statistical comparisons, they are currently absent from the figure.

Reviewer #2 (Remarks to the Author):

The article entitled "Matrigel 3D bioprinting of contractile human skeletal muscle models recapitulating exercise and pharmacological responses" presents the development of a human skeletal muscle model in 24-well-based microplates. In their model, the printing conditions for cell-laden Matrigel have been examined and optimized to study the differentiation and contractions of the model. Furthermore, contractile forces have been measured with known skeletal muscle stimulators to validate the model in terms of exercise and pharmacological responses. The design is quite interesting, and the results are sound. However, it requires some detailed changes as mentioned below.

1. The model proposed by the authors is based on the 3D printing of cell-laden Matrigel that the authors highlight in this work. However, using micro-molding-based techniques seem much easier to reliably and repeatedly generate the desired simple structure. It is required to compare the ways to create the structure and rationalize the specific reasons for selecting 3D printing.

2. Although the authors mentioned that cells do not sediment significantly in Matrigel suspension, it is recommended to show cell density differences in printed constructs in a time-dependent manner.

3. The rationale why the cell density was selected is not mentioned.
4. Representative confocal images for 2D donors are required to compare with the 3D model in Figure 3B.
5. Degradation properties of gels for creating a human skeletal muscle model is also important. Have the authors checked the degradation of Matrigel or compared it with other models using different gel types?

Reviewer #3 (Remarks to the Author):

The paper investigates the bioprinting of mini skeletal muscle tissues in 24 well plate between two anchors for applications in drug testing. The focus is made on the bioink, the electrical pulse stimulation (EPS), the contractile function of the tissues, and the effects of caffeine and tirasemtiv (agonists) on the force developed by the tissues.

The abstract is clear and concise and gives a good overlook about what the paper is. The introduction is correct and appropriate giving the big lines (drug development, tissue model, skeletal muscle tissue, Matrigel as bioink,..). The conclusion is short but appropriate and fits with the results. The paper is correctly written and built. I do not see a particular weakness.

Positive: Developing a microplate for skeletal muscle tissue formation for drug testing applications is challenging. Among the pioneers of this technique is Vandenburg, and this paper is in the continuity and bring new developments. Especially, in the present paper the authors used human cells (primary human skeletal muscle derived cells (hSkMDCs)). They also used pure Matrigel as bioink. This requires to find good parameters for printing (temperature, speed, resolution, Matrigel concentration, good mechanical properties). As notify by the authors usually Matrigel is blended with fibrin (or others) to obtain good printability. Furthermore, the authors highlighted that the use of pure Matrigel is more biomimetic.

Anchoring a skeletal muscle tissue has a marked effect on cell differentiation and myotube formation. However, only few information are available on that particular point. Interestingly, the authors showed that skeletal muscle tissue from a donor of cells aged 40 do not break between posts while from donor aged 19 and aged 17 tissue break faster. The fact that hSkMDCs are used in this study, and that the age of the cell-donor has an impact on the quality and maturity

of the engineered tissue are new information.

Effectively, by using Matrigel bioink, primary human skeletal muscle derived cells, by generating reproducible and homogeneous mini-skeletal muscle tissues in a 24 well plate format, and by using a custom made electrical stimulator system that allows inducing tissue contraction and peak force evaluation, the authors made a noticeable advancement of the technology compared to previous works by Vandenburg and brought new information to the scientific community. Their platform allows significantly mimicking exercise and force increases by known human muscle stimulating drugs.

Negative: One thing which is not fully clear to me is the way the authors used for the posts that anchors each skeletal muscle tissue. The authors mentioned that they put manually a part of pipette tips into the agarose and used them as posts. This is fine to anchor a tissue. However, since figures 5-6 force measurement has been evaluated (from the deflection of the posts), does the fixation of the posts is very solid, and reproducible to generate the same baseline in the force measurement? Does the tips used as posts similar in length and flexibility? Why not using PDMS posts? The authors have used commercial anchors in a previous papers (Laternser 2018 A Novel Microplate 3D Bioprinting Platform for the Engineering of Muscle and Tendon Tissues)

The answer to my question is page 10 in "Materials and methods" where the authors said that "Models were mounted on two hooks that were connected to a force transducer". Maybe the authors should specify earlier and clearly that they use hooks or pipette tips to anchor the tissue, and in what conditions they use one system rather the other.

Comments on Figures:

Fig 1: No special comments. What is the scale bar in c)?

Fig 2: It is clear that anchoring a tissue as an effect on the cell differentiation and myotube

formation, but not so many information are available on this point. The authors made similar observation in their system (hSkMDCs) than other groups with other cell types (like C2C12): Anchoring tissue induced myotubes alignment, higher cell differentiation, contraction in the width of the construct, and possible tissue break depending the flexibility of the posts. However, the authors showed that skeletal muscle tissue from a donor of cells aged 40 do not break between posts while from donor aged 19 and aged 17 tissue break faster, which is a very new information.

Fig 3: The use of specific gene expression and immunostaining to evaluate the cell differentiation over 20 days of culture is correct and the results marked. Interestingly, the authors mentioned that there is cell differentiation into structural and functional myofibers, which (from the Myh pattern) are mainly of embryonal type. This seems to be supported by the images in fluorescence microscopy of Myh and alpha-actinin. Indeed, there is striated patterns but it seems more localized on the borders of the fibers rather than in the center of the fibers, whereas the cell nuclei look to be in the center of the fibers rather than in the borders, both observation support that the myofiber are not mature.

Fig 4: Fig 4a, personally I will put day 12 and day 16 on the side of the fig. not in the middle. Furthermore, since you are using the same Axis (X) titles for both day 12 and day 16, I will close the gap (as much as possible) between the graphs at day 12 and the graphs at day 16. Otherwise, the results of Akt activation and IL-6 induction are clear and marked. This is a good observation of the effect of exercise on skeletal muscle tissue.

Furthermore, the authors showed that their engineered tissue is functional (contractions) and behaves like a natural tissue since the contractions of the tissue are due to physiological stimulations which are blocked by the myosin inhibitor blebbistatin (videos 1 and 2)

Fig 5: In this step the authors optimized the electrical stimulation. Only one type of tissue (19 year old donor) is used, and the tissue is now anchored by 2 hooks connected to a force transducer. They determined the highest stimulation intensity they can subject the tissue (5a), compared single stimulation to multi-stimulations (5b), compared different pulse length (5c), and finally showed the negative effect of the voltage-gated sodium channel blocker tetrodotoxin (TTX) on the contractibility and the peak force production (5d). The results are very good, and their novelty refers more in the use of this cell type (hSkMDCs).

Fig 6: Since the engineered tissue model is well characterized, the authors tested 2 different drugs (agonists) on the engineered skeletal muscle tissue. 10 mM caffeine enhanced the EPS induced peak force and markedly prolonged contractions (AUC) compared to control. 20 μ M troponin C activator Tirasemtiv (CK-2017357) enhanced the EPS induced peak force and markedly prolonged contractions (AUC) compared to control. Furthermore, there is a dose response curve of the effect of Tirasemtiv on peak force. The authors have therefore a reproducible model on which drug testing can be made. This is interesting

Detailed answers (A) to reviewers' comments:

Reviewer #1 (Remarks to the Author):

The paper reports a droplet-on-demand bioprinting protocol of primary-derived human muscle cells in Matrigel. The developed muscle model exhibits high levels of differentiation and is capable of recapitulating many physiological functions, using various immunohistological, electrophysiological and molecular biological experiments to back up the claims.

Q1/C1: *Although the work adds a new aspect for handling pure Matrigel in a bioprinting context, specifically using temperature control and a set concentration range for Matrigel solution bioinks, the physiological constructs made using the droplet-on-demand protocol are not particularly impressive in terms of droplet spatial resolution and overall printed construct size or 3D structures, so the merit of this work in the area of biomaterials is somewhat tenuous. The paper could be improved by investigating the fundamental relations between temperature and Matrigel rheology and improve printing resolution.*

A: The authors thank the reviewer for his/her acknowledgement of the novelty of Matrigel bioprinting and his/her comments and suggestions for investigating temperature-dependence of Matrigel rheology and printing resolution. However, rheological properties of Matrigel as a function of temperature and time have already been described in the mentioned reference 16 (Kleinman & Martin, 2005) in the manuscript on page 11 line 8. To strengthen this, we have added an additional, more recent and more detailed reference 17 by Slater et al. (2017). Regarding the second suggestion of improving Matrigel printing resolution, we think that this goes beyond the scope of this study and is not necessary, as the focus of the paper is the development of a human 3D microphysiological system in the size of a small lower leg mouse muscle such as the EDL muscle (5 mm in length and 1-2 mm in thickness). Our printing resolution, as shown by the printed droplets in Fig 1c, d and resulting 3D models shown in Fig. 1e & 2a, is in our opinion fine for such muscle type.

Q2/C2: *Furthermore, the use of tumorigenic Matrigel as the bioink in this study limits the translational utility of this bioprinting protocol as researchers seek other highly bioactive alternatives that avoid the pitfalls of Matrigel.*

A: The authors agree with the reviewer's comment and a sentence about that has been added in the manuscript at page 4 line 13.

“Despite its tumor origin, which prevents its clinical application, it is still used for many in vitro tissue engineered models because of missing alternatives providing similar biological cues.”

There are also several lapses in writing mechanics in the discussion section and elsewhere that obscure the meaning of the paper. The specific lapses are listed below:

Q3/C3: *Page 12, a comma between “too stiff” and “not allowing” should be present.*

A: The manuscript has been rearranged according to the reviewer's comment on page 13 line 2.

Q4/C4: *Page 17, the sentence beginning with “although the measured...” is unclear and should be rephrased.*

A: The manuscript has been rearranged according to the reviewer's comment on page 17 line 25.

“We did not experience any gelling issues in the microvalves, although the measured temperature at the orifice being 13°C was clearly higher than the 7°C set temperature of the cooling system.”

Q5/C5: *Page 17, the sentence beginning with “although, we did not...” should have the comma between “although” and “we did not...” removed.*

A: The manuscript has been rearranged according to the reviewer's comment on page 18 line 8.

Q6/C6: *Page 18, the sentence beginning with “Drop formation” should have comma between “microvalve” and “before” removed.*

A: The manuscript has been rearranged according to the reviewer's comment on page 18 line 12.

Q7/C7: *Page 18, the sentence beginning with “Furthermore, we demonstrated...” should replace “nor” with “or”.*

A: The manuscript has been rearranged according to the reviewer's comment on page 19 line 7.

In addition to the text itself, the figures describing the results have significant instances of unclarity.

Q8/C8: In Figure 1c, it is not clear what length is represented by the scale bar. Is it 2mm, as indicated at the end of the Figure 1 caption, or something else?

A: The manuscript has been rearranged according to the reviewer's comment and the scale bar of 100 μ m has been added in the legend of Fig. 1C on page 27 line 7.

Q9/C9: In Figure 4, although the caption indicates that there should be some statistical comparisons, they are currently absent from the figure.

A: The manuscript has been rearranged according to the reviewer's comment and the manuscript has been rearranged and the missing statistics were added to Fig. 4.

Reviewer #2 (Remarks to the Author):

The article entitled "Matrigel 3D bioprinting of contractile human skeletal muscle models recapitulating exercise and pharmacological responses" presents the development of a human skeletal muscle model in 24-well-based microplates. In their model, the printing conditions for cell-laden Matrigel have been examined and optimized to study the differentiation and contractions of the model. Furthermore, contractile forces have been measured with known skeletal muscle stimulators to validate the model in terms of exercise and pharmacological responses. The design is quite interesting, and the results are sound. However, it requires some detailed changes as mentioned below.

Q1/C1: *The model proposed by the authors is based on the 3D printing of cell-laden Matrigel that the authors highlight in this work. However, using micro-molding-based techniques seem much easier to reliably and repeatedly generate the desired simple structure. It is required to compare the ways to create the structure and rationalize the specific reasons for selecting 3D printing.*

A: Our ultimate goal is to generate complex tissues as described in our previous paper (Laternser et al., 2018), for which bioprinting is the method of choice. The manuscript has been adapted to better explain this rationale on page 4 lines 21-25. Additionally, we cited a new reference (#24) to underline this point (Jung et al., 2016).

"In our previous work, we printed co-cultures of tenocytes and myoblasts to generate complex muscle-tendon tissue model jetting cells on the top of polymerized bioinks¹⁴. The microvalve-based DOD 3D bioprinting technique allows a controlled spatial deposition of cells, which is not possible with other standard techniques such as micro-molding²⁴. To remain flexible in terms of tissue complexity, we continued using the same bioprinting technique."

Q2/C2: *Although the authors mentioned that cells do not sediment significantly in Matrigel suspension, it is recommended to show cell density differences in printed constructs in a time-dependent manner.*

A: Cell densities in printed droplets were shown in Fig. 1C over 90 min, indicating similar cell densities over this time. The text has been slightly adapted accordingly on page 12 lines 10-12.

"Thus, we did not use the cell stirring system to maintain a homogenous cell suspension in this current project. Indeed, as shown in Figure 1C, single droplets of Matrigel/cell suspensions of comparable size and apparent similar cell content were printed for up to 90 min showing only a 30% decrease in diameter size starting with droplets printed 60 min after cartridge loading (Figure 1D)."

Q3/C3: *The rationale why the cell density was selected is not mentioned.*

A: We added the following sentences at page 18 lines 27-28 and at page 19 lines 1-2 to clarify this:

"We were using the same cell density as previously described by Laternser and colleagues¹⁴. Since muscle tissue is mainly comprised of cells, and has little ECM, we used the highest possible cell density, which doesn't lead to clogging of the microvalve."

Q4/C4: *Representative confocal images for 2D donors are required to compare with the 3D model in Figure 3B.*

A: We performed 2D cell culture experiments with the main donor used for 3D models. Two images have been added to Fig. 3B to allow culture model comparison. Accordingly, the results section and figure legend have been adapted on page 14 lines 9-17 and page 30 lines 1-4, respectively.

"Tissue differentiation and architecture of the models was further analyzed by histology and compared to standard 2D cultures (Figure 3B). Whole-mount immunostaining of 3D models for Myh showed multinucleated, aligned striated myofibers indicating sarcomeric organization already at day 6 of differentiation where 2D cultures showed less densely packed and aligned fibers with only weak Myh-staining. At day 10 of differentiation, 2D cultures remained as loosely packed myofibers, albeit with stronger Myh-expression, whereas 3D models showed fully compacted and aligned myofibers. Furthermore, immunostaining for muscle-specific actin-anchoring α -actinin and staining for filamentous actin (F-actin) confirmed the myofibers alignment and striation in 3D models."

“Histological analyses of tissue cultures of a 17-year old donor. Left image panel: Confocal immunofluorescence microscopy of Myosin heavy chain (Myh) expression in control 2D cultures of differentiation day 6 and 10 as indicated in the figure.”

Q5/C5: Degradation properties of gels for creating a human skeletal muscle model is also important. Have the authors checked the degradation of Matrigel or compared it with other models using different gel types?

A: Matrigel is a natural ECM extract whose main components (laminin, collagen IV) are found in basal lamina of muscle tissue. Indeed, Matrigel has been shown to stimulate muscle tissue differentiation in vitro (Grefte et al., 2012). Thus, we did not check for Matrigel degradation. The reference (#18) has been added on page 4 line 13.

“Matrigel is a commonly used extract of extracellular basement membrane proteins derived from the Engelbreth-Holm-Swarm (EHS) mouse sarcoma to support growth and differentiation of cells in culture ¹⁶⁻¹⁸»

Reviewer #3 (Remarks to the Author):

The paper investigates the bioprinting of mini skeletal muscle tissues in 24 well plate between two anchors for applications in drug testing. The focus is made on the bioink, the electrical pulse stimulation (EPS), the contractile function of the tissues, and the effects of caffeine and tirasemtiv (agonists) on the force developed by the tissues.

The abstract is clear and concise and gives a good overlook about what the paper is. The introduction is correct and appropriate giving the big lines (drug development, tissue model, skeletal muscle tissue, Matrigel as bioink,...). The conclusion is short but appropriate and fits with the results. The paper is correctly written and built. I do not see a particular weakness.

Positive: Developing a microplate for skeletal muscle tissue formation for drug testing applications is challenging. Among the pioneers of this technique is Vandenburg, and this paper is in the continuity and bring new developments. Especially, in the present paper the authors used human cells (primary human skeletal muscle derived cells (hSkMDCs)). They also used pure Matrigel as bioink. This requires to find good parameters for printing (temperature, speed, resolution, Matrigel concentration, good mechanical properties). As notify by the authors usually Matrigel is blended with fibrin (or others) to obtain good printability. Furthermore, the authors highlighted that the use of pure Matrigel is more biomimetic.

Anchoring a skeletal muscle tissue has a marked effect on cell differentiation and myotube formation. However, only few information are available on that particular point. Interestingly, the authors showed that skeletal muscle tissue from a donor of cells aged 40 do not break between posts while from donor aged 19 and aged 17 tissue break faster. The fact that hSkMDCs are used in this study, and that the age of the cell-donor has an impact on the quality and maturity of the engineered tissue are new information.

Effectively, by using Matrigel bioink, primary human skeletal muscle derived cells, by generating reproducible and homogeneous mini-skeletal muscle tissues in a 24 well plate format, and by using a custom made electrical stimulator system that allows inducing tissue contraction and peak force evaluation, the authors made a noticeable advancement of the technology compared to previous works by Vandenburg and brought new information to the scientific community. Their platform allows significantly mimicking exercise and force increases by

known human muscle stimulating drugs.

Negative: One thing which is not fully clear to me is the way the authors used for the posts that anchors each skeletal muscle tissue.

Q1/C1: The authors mentioned that they put manually a part of pipette tips into the agarose and used them as posts. This is fine to anchor a tissue. However, since figures 5-6 force measurement has been evaluated (from the deflection of the posts), does the fixation of the posts is very solid, and reproducible to generate the same baseline in the force measurement?

A: Contractile force measurements were not done by post deflection but were done directly in a tissue chamber hooking-up the models to force transducers as described in M&Ms on page 10 and in Results on page 15.

Q2/C2: Does the tips used as posts similar in length and flexibility? Why not using PDMS posts?

A: As written in the M&Ms section all the tips were manually cut at the same position to obtain posts with similar structure, flexibility, diameter, and length. They were only used for tissue maturation and not for force measurements.

We didn't want to use PDMS posts because of their known limitations for drug testing applications as they exhibit high absorption of lipophilic molecules. To clarify this, we inserted a text on PDMS on page 18 lines 21-25 and added a reference (#38) (Sano et al, 2019).

“Vandenburgh et al.¹¹ used flexible posts made of polydimethylsiloxane (PDMS) to generate differentiated muscle tissues. PDMS is known for its ease of use and processing and is thus frequently employed as a biocompatible polymer in micro-fluidic device development. Unfortunately, the material exhibits strong absorption of small hydrophobic molecules and makes it unsuitable in combination with drug testing³⁸.”

Q3/C3: The authors have used commercial anchors in a previous papers (Latenser 2018 A Novel Microplate 3D Bioprinting Platform for the Engineering of Muscle and Tendon Tissues)

The answer to my question is page 10 in “Materials and methods” where the authors said that “Models were mounted on two hooks that were connected to a force transducer”. Maybe the authors should specify earlier and clearly that they use hooks or pipette tips to anchor the tissue, and in what conditions they use one system rather the other.

A: Same answer as in Q1/C1. Contractile force measurements were not done by post deflection but were done directly in a tissue chamber hooking-up the models to force transducers as described in M&Ms on page 10 and in Results on page 15.

Comments on Figures:

Q4/C4: Fig 1: No special comments. What is the scale bar in c)?

A: The manuscript has been rearranged and the scale bar of 100 μm has been added in the legend Fig. 1C on page 27 line 7.

Q5/C5: Fig 2: It is clear that anchoring a tissue as an effect on the cell differentiation and myotube formation, but not so many information are available on this point. The authors made similar observation in their system (hSkMDCs) than other groups with other cell types (like C2C12): Anchoring tissue induced myotubes alignment, higher cell differentiation, contraction in the width of the construct, and possible tissue break depending the flexibility of the posts. However, the authors showed that skeletal muscle tissue from a donor of cells aged 40 do not break between posts while from donor aged 19 and aged 17 tissue break faster, which is a very new information.

A: Although, we saw differences in model stability over time depending on the donor age, we cannot draw an age-correlation at this point since we did not test enough donors. These differences are just donor specific. We have edited the corresponding text in red on page 19 lines 25-26.

“In any case, cell culture inserts with micro-posts of defined variable stiffness and models from many more donors at different age will be required to analyze the relationships between post deflection with respect to stiffness, tissue differentiation, model rupture and different donor age”

Q6/C6: Fig 3: The use of specific gene expression and immunostaining to evaluate the cell differentiation over 20 days of culture is correct and the results marked. Interestingly, the authors mentioned that there is cell differentiation into structural and functional myofibers, which (from

the Myh pattern) are mainly of embryonal type. This seems to be supported by the images in fluorescence microscopy of Myh and alpha-actinin. Indeed, there is striated patterns but it seems more localized on the borders of the fibers rather than in the center of the fibers, whereas the cell nuclei look to be in the center of the fibers rather than in the borders, both observation support that the myofiber are not mature.

A: We agree that myofibers are not mature but rather embryonal slow-type muscle fibers based on Myh gene expression pattern. The text has been adapted accordingly on page 19 lines 22-23.

“Nevertheless, according to the Myh expression pattern the in vitro engineered 3D models from both donors mainly consisted of slow and embryonal type myofibers.”

Q7/C7: Fig 4: Fig 4a, personally I will put day 12 and day 16 on the side of the fig. not in the middle. Furthermore, since you are using the same Axis (X) titles for both day 12 and day 16, I will close the gap (as much as possible) between the graphs at day 12 and the graphs at day 16. Otherwise, the results of Akt activation and IL-6 induction are clear and marked. This is a good observation of the effect of exercise on skeletal muscle tissue.

Furthermore, the authors showed that their engineered tissue is functional (contractions) and behaves like a natural tissue since the contractions of the tissue are due to physiological stimulations which are blocked by the myosin inhibitor blebbistatin (videos 1 and 2)

A: Thank you for your comments. We have changed the picture 4a) layout, accordingly, thus improving the clarity.

Q8/C8: Fig 5: In this step the authors optimized the electrical stimulation. Only one type of tissue (19 year old donor) is used, and the tissue is now anchored by 2 hooks connected to a force transducer. They determined the highest stimulation intensity they can subject the tissue (5a), compared single stimulation to multi-stimulations (5b), compared different pulse length (5c), and finally showed the negative effect of the voltage-gated sodium channel blocker tetrodotoxin (TTX) on the contractibility and the peak force production (5d). The results are very good, and their novelty refers more in the use of this cell type (hSkMDCs).

A: Thank you for this comment. Yes, indeed, these are new results for this human cell type in combination with a read-out system, which is described in our manuscript.

Q9/C9: *Fig 6: Since the engineered tissue model is well characterized, the authors tested 2 different drugs (agonists) on the engineered skeletal muscle tissue. 10 mM caffeine enhanced the EPS induced peak force and markedly prolonged contractions (AUC) compared to control. 20 μ M troponin C activator Tirasemtiv (CK-2017357) enhanced the EPS induced peak force and markedly prolonged contractions (AUC) compared to control. Furthermore, there is a dose response curve of the effect of Tirasemtiv on peak force. The authors have therefore a reproducible model on which drug testing can be made. This is interesting*

A: We fully agree with the reviewer's comment and we think that this publication will have a great impact in the field.

REVIEWERS' COMMENTS:

Reviewer #1 (Remarks to the Author):

the manuscript is improved after revision and suitable for publication.

Reviewer #2 (Remarks to the Author):

The authors have addressed nearly all of my concerns in their revisions. I am satisfied with the revised manuscript.

Reviewer #3 (Remarks to the Author):

For Communications Biology

Review of revised manuscript number COMMSBIO-20-3456A entitled "Matrigel 3D bioprinting of contractile human skeletal muscle models recapitulating exercise and pharmacological responses"

The authors added the needed details to the manuscript and answered to my questions. I am fully satisfied with the present manuscript.

Moreover, the information added to the manuscript to answer the comments of others reviewers are interesting, precise, and improve the paper too.

This is a good paper.

Misspells:

(μ N) in the Y axis Fig 1 d, / Fig 5 b, c, d / Fig 6 d